

# Implications of high-dose vitamin D₃ with and without vitamin C on bone mineralization and blood biochemical factors in broiler breeder hens and their offspring

Ruhollah Kianfar[1], Reza Kanani[1], Hossein Janmohammadi[1], Majid Olyaee[1], Maghsoud Besharati[2] and Maximilian Lackner[3]

[1] Department of Animal Sciences, Faculty of Agriculture, University of Tabriz, Tabriz, Iran
[2] Department of Animal Science, Ahar Faculty of Agriculture and Natural Resources, University of Tabriz, Tabriz, Iran
[3] Department of Industrial Engineering, University of Applied Sciences Technikum Wien, Vienna, Austria

Corresponding authors
Ruhollah Kianfar,
Rkianfar@tabrizu.ac.ir
Maximilian Lackner,
maximilian.lackner@technikum-wien.at

## ABSTRACT

As broiler breeder hens age, they often experience a decline in bone mineralization and calcium absorption, especially during the later stages of egg production. This issue not only affects the hens' health, making them more prone to conditions like osteoporosis, but it also impacts the quality of their offspring. To tackle this problem, our study explores whether supplementing these hens with a combination of vitamins D₃ and C could help improve their bone health and overall biochemical balance, both for them and their progeny. The goal of this research was to evaluate the effects of high doses of vitamin D₃, with and without added vitamin C, on bone mineralization and key blood parameters in aging broiler breeder hens and their offspring. In this experiment, 240 hens and 24 roosters from the Ross 308 strain, aged between 49 and 61 weeks, were used, and a two-way ANOVA ($2 \times 2$) design was applied. This involved two levels of vitamin D₃ (3,500 IU and 5,500 IU) and two levels of vitamin C (0 and 150 mg/kg), with six replications of 10 hens and one rooster per group. At the end of the study, blood samples were collected from hens and their offspring for biochemical analysis, and tibia bones were taken for ash content and mineralization assessment. The findings showed that vitamin D₃ supplementation significantly lowered blood cholesterol, alkaline phosphatase (ALP), and parathyroid hormone (PTH) levels ($P < 0.05$), while boosting calcium, 25-hydroxycholecalciferol (25(OH)D₃), and 1,25-dihydroxycholecalciferol (1,25(OH)₂D₃) ($P < 0.05$). Higher doses of vitamin D₃ also improved the strength, resistance, and ash content of the hens' tibia bones, and increased calcium in the carcasses of their offspring. Adding 150 mg/kg of vitamin C to the diet also had a positive effect, reducing cholesterol, ALP, and PTH, while enhancing plasma calcium, total antioxidant capacity, and the active form of vitamin D₃ ($P < 0.05$). Vitamin C supplementation significantly strengthened the tibial bones of the hens and improved plasma calcium and PTH levels in their offspring ($P < 0.05$). Interestingly, combining elevated doses of both vitamins D₃ and C resulted in even greater improvements in tibial bone strength ($P < 0.05$). In conclusion, giving hens 150 mg of vitamin C along with 5,500 IU of vitamin D₃ leads to substantial improvements in the calcium content

and structural integrity of their bones, and also boosts calcium and ash content in the carcasses of their offspring.

## INTRODUCTION

As broiler breeder hens age, keeping their bones strong becomes increasingly difficult, especially during the intense periods of egg production. With age, their ability to absorb calcium diminishes, forcing them to draw calcium from their medullary bones to meet the demands of egg production. This depletion raises the risk of osteoporosis, which not only jeopardizes the hens' health but also impacts their reproductive performance and the quality of their offspring. The severity of this issue has been well-documented, with researchers like *Kim et al. (2007)* highlighting the urgent need for effective interventions. Given these challenges, it is crucial to improve the diet of these aging hens by supplementing with vitamins $D_3$ and C to support optimal calcium absorption, especially during the later stages of the laying cycle. Vitamin $D_3$ (cholecalciferol) plays a key role in calcium metabolism by promoting the active transport of calcium through the intestinal wall. However, the exact process by which vitamin $D_3$ enhances calcium absorption is still not fully understood. Recent studies, including research by *Fatemi et al. (2024)*, have found that vitamin $D_3$ promotes the synthesis of a protein in the intestinal lining, which is essential for calcium transport. Without this protein, the absorption of calcium and phosphorus is significantly reduced, leading to their accumulation in insoluble forms within the intestines. This makes adequate vitamin D levels critical for maintaining the body's ability to absorb calcium and phosphorus efficiently. Once synthesized, $1,25(OH)_2D_3$ enters the nuclei of intestinal cells and interacts with genetic material to produce specialized RNA (*Fakhoury et al., 2020*). These RNAs encode calcium-binding proteins (CaBP), which are vital for transporting calcium and improving bone mineralization (*Leyva-Jimenez et al., 2019*). *Goodson-Williams, Roland Sr & McGuire (1986)* demonstrated that increasing dietary vitamin $D_3$ supports the formation of medullary bone and reduces the breakdown of cortical bone in older hens. This is further backed by studies from *Peng et al. (2013)* and *Atencio, Edwards & Pesti (2005a)*, who showed that adding vitamin $D_3$ to the diet significantly improves bone health and mineralization in broiler breeder hens and their offspring. In addition to vitamin $D_3$, vitamin C also plays an essential role. Known for its antioxidant properties, vitamin C combats oxidative stress by neutralizing free radicals. Additionally, it enhances nutrient absorption by mitigating the negative effects of corticosteroid hormones, which can impair the body's ability to utilize nutrients (*McDowell & Cunha, 2012*).

Vitamin C plays a multifaceted role in maintaining bone health. It not only promotes collagen synthesis, which is vital for bone formation, but also supports the activation of enzymes involved in vitamin D metabolism (*DePhillipo et al., 2018*; *Aghajanian et al.,*

*2015*). Recent research suggests that vitamin C influences the expression of genes in osteoblasts, as well as the production of bone matrix proteins and growth factors necessary for bone metabolism (*Hasan et al., 2024*). Although previous studies have shown that supplementing vitamin C improves bone strength in broiler chickens (*Lohakare, Chae & Hahn, 2004*), its precise impact on vitamin $D_3$ metabolism and calcium absorption remains somewhat unclear. One of the critical actions of vitamin C is its role in converting $25(OH)D_3$ into its active form, which is essential for maximizing calcium absorption in the intestines and renal tubules (*Van Driel & Van Leeuwen, 2023*). Despite various studies pointing to the benefits of vitamins $D_3$ and C on bone health and poultry performance, there is still limited understanding of how these vitamins interact and what the optimal dosing strategies are. For instance, while *Reyes et al. (2021)* found that different levels of dietary vitamin C improved overall performance in laying hens, they did not observe any significant changes in egg quality or bone structure. This knowledge gap is significant. Aging broiler breeder hens experience declining bone health, which not only impacts their productivity but also affects the quality of their offspring and overall flock welfare. Understanding the interactions between vitamins $D_3$ and C, and determining the best dosage for supplementation, could offer valuable guidelines for enhancing bone health and production efficiency in poultry farming. Given the decline in calcium absorption during the later stages of egg production and the vital roles that vitamins $D_3$ and C play in boosting this absorption, our study aims to investigate the effects of high-dose vitamin $D_3$, both with and without vitamin C supplementation, on bone mineralization and key blood biochemical markers in broiler breeder hens and their offspring. By examining these interactions, we hope to provide practical insights into nutritional strategies that could help mitigate the adverse effects of aging on bone health in poultry, ultimately improving their productivity and well-being.

## MATERIALS AND METHODS

### Preparation and management

This experiment involved 240 hens and 24 roosters from the Ross 308 strain, aged between 49 and 61 weeks. We used a two-way ANOVA ($2 \times 2$) design to test two levels of vitamin $D_3$ (3,500 IU and 5,500 IU) and two levels of vitamin C (0 and 150 mg/kg), following a completely randomized setup with six replications. The level of vitamin $D_3$ was selected based on the work of *Wideman Jr et al. (2015)*, *Atencio, Edwards & Pesti (2005a)* and *Atencio, Pesti & Edwards Jr (2005b)*. Each replication group included 10 hens and one rooster, as described by *Kanani et al. (2023)*. The birds were housed in experimental floor pens, each measuring $1.5 \times 3$ m. To prepare the diets, a vitamin $D_3$ premix containing five million IU/kg (Darusazan Iran Co, Tehran, Iran) and a vitamin C premix with 150 mg/kg (Darusazan Iran Co, Tehran, Iran) were used, combined with a basal diet formulated according to Ross 308 guidelines (*Aviagen, 2018*). The experimental diets maintained consistent energy and protein levels (as outlined in Table 1). Birds were kept at an average temperature of 24 °C, with unlimited access to water and a controlled amount of feed. A 14-hour light and 10-hour dark cycle was provided using LED lamps. To avoid cross-contamination, feed was separated between the hens and roosters, with distinct feeders

**Table 1  Basal diet ingredients and chemical composition.**

| Dietary ingredient | (%) |
| --- | --- |
| Corn grain | 60.38 |
| Soybean meal (44%) | 16.58 |
| Barley grain | 5.00 |
| Vegetable oil | 2 |
| Wheat bran | 4.75 |
| Di Ca–Phosphate (17%) | 1.09 |
| Oyster shell | 5.58 |
| $CaCo_3$ | 3.02 |
| NaCl | 0.22 |
| $NaHCO_3$ | 0.41 |
| Cholin chloride | 0.050 |
| DL-Methionine | 0.18 |
| L-Lysine HCl | 0.03 |
| L-Threonine | 0.01 |
| Vitamin and mineral premix[1] | 0.5 |
| Inert filer (sand) | 0.200 |
| Calculated chemical composition (%) | |
| AMEn (Kcal/kg)[2] | 2800 |
| Crude protein | 13 |
| Calcium | 3.4 |
| Av.phosphorus | 0.32 |
| Na | 0.23 |
| Methionine+cystine | 0.65 |
| Methionine | 0.41 |
| Lysine | 0.72 |
| Threonine | 0.53 |
| DCAB (meq/kg)[3] | 210 |

**Notes.**

Vitamin and mineral premix: vitamin A, 12,200 IU; vitamin $D_3$, 0 IU; Menadione, 2.9 mg; vitamin E, 100 IU; Thiamin, 3.2 mg; Riboflavin, 12.6 mg; pyridoxine, 5.5 mg; cobalamin, 0.046 mg; vitamin B5, 20 mg; vitamin B3, 40 mg; folic acid, four mg; biotin, 0.29 mg; Se, 0.29 mg; CU, 20 mg; Fe, 50 mg; Mn, 115 mg; Zn, 100 mg; Iodine, 2 mg; AMEn, Apparent metabolizable energy, N-corrected, poultry; DCAB, Dietary Cation Anion Balance.
Difference between the gross energy in the feed and the gross energy in the feces, urines and gasses, corrected for a nitrogen balance of 0.

for each. Each pen was equipped with two nipple drinkers and laying nests for manual egg collection. Before starting the experiment, each bird was weighed individually and assigned to its respective pen. The chickens were sourced from Almas Chicken Commercial Poultry Farm located in the city of in Tabriz, and all husbandry practices followed the Ross 308 commercial strain manual to ensure proper care. At the end of the experiment, the hens, roosters, and day-old chicks were euthanized humanely using cervical dislocation, a standard practice in poultry processing. Since no toxic or medicinal substances were used during the experiment, no special carcass disposal protocols were necessary. Carcasses were incinerated in the farm's commercial incinerator, while any remaining live birds were sent to a slaughterhouse for industrial meat powder production.

## Measurement of blood biochemical parameters of broiler breeder hens

At the 12th week of the experiment, two hens from each group were randomly selected for blood collection *via* the wing vein. The blood samples were then centrifuged at 3,000 rpm for 15 min to separate the serum, which was stored in 1.5 mL microtubes for further analysis. Total antioxidant capacity (TAC), total cholesterol, and triglycerides were measured using analytical kits from Pars Azmon, and the readings were taken with a 617-CLIMA device (RALCO, Catalonia, Spain) following standard photometric methods. Plasma glucose levels were determined using the Trinder method (*1969*) with diagnostic kits from Diamond Diagnostics (Holliston, MA, USA). Calcium levels were assessed using an autoanalyzer, while estrogen and progesterone concentrations were measured using radioimmunoassay (RIA) with DSL-43100 and DSL-3900 kits from Diagnostic Systems Laboratories Inc. (Webster, TX, USA) according to Abraham's protocol (*Abraham, 1977*).

Serum parathyroid hormone (PTH) was assessed using a chicken parathyroid hormone ELISA test kit with the code NBE-234985 (NebuEasy™, China). The concentration of calcitonin was evaluated using an ELISA test kit with the code NBE-234987 (NebuEasy™, China). Serum 25-OH-$D_3$ was determined using a 25-Hydroxyvitamin $D_3$ ELISA kit from PadtanGostar, Iran. Serum $1,25(OH)_2$ vitamin $D_3$ measurement was performed using an EIA kit (IDS Ltd, Tyne & Wear, UK), where the first step involved immunoextraction, followed by quantification using enzyme immunoassay. Liver enzymes, including alanine aminotransferase (ALT), aspartate aminotransferase (AST), and alkaline phosphatase (ALP), were also analyzed using commercial kits (Pars Azmon, Iran) with an auto analyzer.

## Bone mineral density of the tibia bone in broiler breeder hens

At the end of the 12-week study, two hens from each group were euthanized, and their left and right tibia bones were carefully collected for analysis. The left tibia was used to assess breaking strength, while the right tibia underwent analysis for ash, calcium, and phosphorus content. To prepare the right tibia for analysis, any surrounding tissue was first removed and then the bone was boiled in water for one minute. After boiling, the bone was immersed in ethyl ether for 24 h to eliminate any residual fat. Once cleaned, the bones were dried at 65 °C for 24 h. After drying, they were weighed and subjected to further analysis by heating at 550 °C to determine the bone's ash content, following methods outlined by *Zhang et al. (2021)*. The concentrations of calcium and phosphorus in the bones were quantified using an atomic absorption spectrophotometer, in accordance with *AOAC International (2005)* standards. The left tibia was also dried at 65 °C for 72 h and then evaluated for fracture strength. A perpendicular force was applied at a rate of five mm/min using a cylindrical knife (30 mm diameter) positioned at the midsection between the epiphyses, with an EMIC DL2000 INSTRON device from Paraná, Brazil. The maximum force required to fracture the bone was recorded for analysis.

## Measurement of blood biochemical parameters and mineral substances of the broiler breeder hens offspring

At the end of the experiment, fertile eggs from all replicates were transferred to the incubator on the same day to ensure uniform hatching conditions. Chicks were hatched within a single

hatching period and processed immediately after hatching to minimize variability. For each replicate, four chicks were randomly selected for blood sampling, and an additional four chicks from the same hatching were used for mineral analysis to avoid any potential effects of blood collection on carcass ash and mineral content. Blood samples were collected from the heart of the chicks using sterile 25-gauge needles, with approximately three ml of blood taken from each chick for serum separation. Plasma levels of calcium, $25(OH)D_3$, $1,25(OH)_2D_3$, PTH, ALP, and phosphorus were measured. For mineral analysis, these four chicks were euthanized using cervical dislocation. The entire carcass of each chick was ground and dried at 65 °C for 72 h. Post-weighing, carcass ash, calcium, and phosphorus content were determined after heating at 550 °C (*AOAC International, 2005*), with calcium and phosphorus assessed using an atomic absorption device.

### Statistical analysis

The data in this study were obtained from a factorial experiment involving two factors, designed as a completely randomized design. A two-way analysis of variance (ANOVA) was employed, incorporating six replications. All statistical analyses were conducted SAS 9.4 software using (*SAS Institute, 2019*). Significant differences among trait means were assessed using the Tukey-Kramer method.

### Ethical statement

The study adhered to ethical guidelines for animal research, ensuring humane treatment throughout the experiment, including during the euthanasia process. All handling and management of the poultry complied with established welfare protocols and relevant breeding management guidelines. Additionally, the project conformed to the ethical principles and national standards for conducting medical research in Iran (approval reference number: IR.TABRIZU.REC.1403.008; Research Ethic Committees of University of Tabriz).

## RESULTS AND DISCUSSION

### Effect of vitamin $D_3$ and vitamin C on mineralization of tibia bones in broiler breeder hens

The effects of vitamin $D_3$ and vitamin C on the dry matter, ash, calcium, phosphorus content, and resistance to fracture of the tibia bones in broiler breeder hens are summarized in Table 2. The study revealed significant effects of vitamin $D_3$ on the resistance ash and phosphorous content of tibia bones in broiler breeder hens ($P < 0.05$). Specifically, higher doses of vitamin $D_3$ (5,500 IU) led to a notable increase in tibia ash percentage and bone resistance compared to the lower dose of 3,500 IU ($P < 0.05$). This finding is consistent with the work of *Atencio, Edwards & Pesti (2005a)*, who also reported that dietary supplementation of 4,000 IU/kg of vitamin $D_3$ significantly boosted bone ash concentration in broiler breeder hens.

Interestingly, while higher tibial calcium levels were numerically observed in the group receiving the higher vitamin $D_3$ dose, significant differences in dry matter, calcium, or phosphorus content in the tibia were not shown by statistical analysis. Previous research

**Table 2** The effects of vitamin $D_3$ and vitamin C on the dry matter, ash, calcium, phosphorus content, and resistance to fracture of the tibia bones in broiler breeder hens.

| Treatments | | Tibia DM (%) | Tibia ASH (%) | Tibia Ca (g/Kg) | Tibia p (g/Kg) | Tibia resistance (N) |
|---|---|---|---|---|---|---|
| $D_3$ | 3,500 IU | 74.12 | 54.04[b] | 374.41 | 171.58 | 295.58[b] |
| | 5,500 IU | 75.51 | 55.76[a] | 379.50 | 168.08 | 356.80[a] |
| *P-Value* | | 0.137 | 0.0001 | 0.211 | 0.185 | 0.001 |
| SEM | | 0.633 | 0.221 | 2.78 | 1.80 | 9.57 |
| Vitamin C | 0 mg | 74.68 | 54.45[b] | 376.16 | 169.66 | 310.51[b] |
| | 150 mg | 74.94 | 55.35[a] | 377.75 | 170.00 | 341.87[a] |
| *P-Value* | | 0.776 | 0.010 | 0.692 | 0.897 | 0.049 |
| SEM | | 0.633 | 0.221 | 2.78 | 1.80 | 9.57 |
| Interaction | | | | | | |
| 3,500 $D_3 \times$ 0 C | | 73.95 | 53.74 | 373.50 | 170.50 | 261.59[b] |
| 3,500 $D_3 \times$ 150 C | | 74.29 | 54.35 | 375.33 | 172.66 | 329.56[a] |
| 5,500 $D_3 \times$ 0 C | | 75.42 | 55.17 | 378.83 | 168.83 | 359.42[a] |
| 5,500 $D_3 \times$ 150 C | | 75.59 | 56.34 | 380.16 | 167.33 | 354.17[a] |
| SEM | | 0.313 | 0.896 | 3.94 | 2.55 | 13.54 |
| *P-Value* | | 0.923 | 0.378 | 0.950 | 0.481 | 0.026 |

**Notes.**
[a-b]Means with different superscripts within a column differ significantly ($P < 0.05$).
SEM, Standard Error of the Mean; DM, Dry Matter.

supports this nuance; for example, *Atencio, Edwards & Pesti (2005a)* demonstrated that including 25-hydroxyvitamin $D_3$ can significantly enhance tibia calcium and strength in laying hens at specific ages. However, *Li et al. (2021)* found that different levels of vitamin $D_3$ and its metabolites did not significantly influence the lean bone index, tibial ash, or phosphorus content, indicating a complex relationship between vitamin $D_3$ levels and skeletal development. Vitamin $D_3$ is crucial for facilitating calcium deposition in bones, thereby improving their quality. Recent findings by *Chen et al. (2020)* suggest that $25(OH)D_3$ can encourage bone growth during the early stages of development, resulting in better mineral deposition later on. Enhancing bone quality during these developmental phases is essential for optimizing skeletal robustness in broilers, as highlighted by *Li et al. (2021)*.

The effects of vitamin C on the resistance to fracture and ash content of the tibia bones in broiler breeder hens were significant ($P < 0.05$). Supplementation with vitamin C at a dose of 150 mg significantly improved the tibia ash percentage and resistance compared to the control group that did not receive vitamin C. While no other traits showed significant enhancements from vitamin C supplementation, the mechanism by which it promotes bone mineralization is quite interesting. Vitamin C stimulates the enzyme 25-hydroxycholecalciferol hydroxylase, which is essential for converting 25-hydroxyvitamin $D_3$ ($25(OH)D_3$) into its active form, $1,25(OH)_2D_3$. This active metabolite enhances calcium absorption from the intestinal lumen, thereby positively impacting bone mineralization (*Shi et al., 2024*).

Furthermore, research by *Chung et al. (2005)* indicated that vitamin C supplementation at 200 mg/kg can help alleviate the adverse effects of high ambient temperatures, resulting in improved tibial strength. Our study echoed these findings, showing that hens fed a diet enriched with vitamin C exhibited the highest tibial fracture strength, demonstrating significant differences compared to both the control and vitamin E groups. Similarly, *Orban et al. (1993)* reported increased fracture strength in the femurs of broilers supplemented with ascorbic acid.

Notably, significant interactions between vitamin $D_3$ and vitamin C were observed regarding resistance to fracture. The combination of 5,500 IU of vitamin $D_3$ with 150 mg of vitamin C resulted in the best tibia resistance, while the lowest resistance values were recorded in the group receiving 3,500 IU of vitamin $D_3$ without vitamin C. However, despite these interactions, the different doses of vitamin $D_3$ and vitamin C did not significantly change other parameters. Overall, our findings suggest that both vitamin $D_3$ and vitamin C have a positive influence on tibial mineralization, bone ash, and strength in broiler breeder hens, although the most pronounced effects were observed when these vitamins were administered separately rather than in combination. This result, confirmed by *Lohakare et al. (2005)* concluded that supplementing vitamin C with vitamin D improves bone strength through their interactive effects. Vitamin C, crucial for collagen synthesis, acts as a cofactor for enzymes involved in converting proline and lysine into hydroxyproline and hydroxylysine key steps in forming the collagen fibril network required for bone mineralization. This process, reliant on hydroxyapatite deposition, is influenced by active metabolites of vitamin D, emphasizing the synergistic roles of both vitamins in bone metabolism (*Tillman, 1993*). Additionally, vitamin C enhances calcium-binding protein capacity, potentially impacting calcium metabolism (*Orban et al., 1993*).

## Effect of vitamin $D_3$ and vitamin C on mineralization of broiler breeder offspring

The effect of vitamin $D_3$ and vitamin C on the dry matter, ash, calcium, and phosphorus content of broiler breeder offspring carcasses are summarized in Table 3. Our findings reveal that vitamin $D_3$ had a significant impact on the dry matter and calcium content of the offspring ($P < 0.05$). Specifically, higher levels of vitamin $D_3$ were associated with increased calcium percentages in the offspring compared to those receiving lower doses. However, it is noteworthy that the administration of vitamin $D_3$ did not significantly affect other measured parameters.

Previous research, including the work by *Atencio, Edwards & Pesti (2005a)*, supports the crucial role of vitamin $D_3$ in enhancing bone mineralization in the progeny of broiler breeder hens. They found that incorporating 4,000 IU/kg of vitamin $D_3$ into hen diets led to higher tibia bone ash and a reduction in the prevalence of rickets and tibia dyschondroplasia in the offspring, which contrasts with some findings observed in our study.

Additionally, maternal 25-hydroxyvitamin $D_3$ has been shown to provide protective benefits to the developing embryo, decreasing early embryonic mortality and positively influencing the bone mineral density of the offspring (*Saunders-Blades & Korver, 2014*). *Khan et al. (2010)* demonstrated that feeding broiler chicks diets containing 200 to 3,000

**Table 3** The effect of vitamin $D_3$ and vitamin C on the dry matter, ash, calcium, and phosphorus content of broiler breeder offspring carcasses.

| Treatments | | DM (%) | ASH (%) | Ca (%) | p (%) |
|---|---|---|---|---|---|
| $D_3$ | 3,500 IU | 28.80[a] | 7.34 | 15.24[b] | 11.37 |
| | 5,500 IU | 27.04[b] | 7.91 | 16.71[a] | 11.85 |
| *P-Value* | | 0.0001 | 0.066 | 0.002 | 0.355 |
| SEM | | 0.206 | 0.209 | 0.307 | 0.360 |
| Vitamin C | 0 mg | 27.76 | 7.43 | 15.16[b] | 11.11 |
| | 150 mg | 28.08 | 7.81 | 16.76[a] | 12.11 |
| *P-Value* | | 0.295 | 0.129 | 0.001 | 0.064 |
| SEM | | 0.206 | 0.209 | 0.307 | 0.360 |
| Interaction | | | | | |
| 3,500 $D_3 \times$ 0 C | | 27.29[b] | 7.18 | 14.75 | 10.98 |
| 3,500 $D_3 \times$ 150 C | | 30.32[a] | 7.50 | 15.73 | 11.76 |
| 5,500 $D_3 \times$ 0 C | | 28.24[b] | 7.69 | 15.58 | 11.24 |
| 5,500 $D_3 \times$ 150 C | | 25.84[c] | 8.13 | 17.85 | 12.46 |
| SEM | | 0.291 | 0.295 | 0.434 | 0.510 |
| *P-Value* | | 0.0001 | 0.841 | 0.155 | 0.669 |

**Notes.**
[a-c] Means with different superscripts within a column differ significantly ($P < 0.05$).
SEM, Standard Error of the Mean; DM, Dry Matter; Ca, Calcium; p, Phosphorus.

IU/kg of vitamin $D_3$ from hatching to 42 days significantly increased tibia and toe ash levels. This underscores the importance of optimal vitamin $D_3$ levels for regulating calcium and phosphorus metabolism, which are essential for healthy bone development, as highlighted by *Sinclair-Black, Garcia & Ellestad (2023)*.

*Ameenuddin et al. (1986)* also reported that administering 5,000 IU of vitamin $D_3$ to hens notably raised the tibia bone ash of their offspring. Supporting this, *Atencio, Pesti & Edwards Jr (2005b)* found that varying doses of vitamin $D_3$ positively impacted the ash content of tibia bones in progeny, reinforcing the vitamin's critical role in maintaining skeletal integrity. When it comes to vitamin C, the primary effect observed was a notable increase in calcium content in the offspring of broiler breeder hens ($P < 0.05$), though no significant changes were detected in other traits. Research has long highlighted vitamin C's essential role in bone development, particularly through its promotion of hydroxyproline production, a critical component of collagen. *Farquharson et al. (1998)* suggested that ascorbic acid enhances chondrocyte matrix production and stimulates the synthesis of $1,25(OH)_2D_3$, which is associated with an upregulation of vitamin D receptors. These collagen fibrils are vital for maintaining proper bone structure and development (*Revell et al., 2021*).

In times of stress, the body's natural production of vitamin C may fall short of physiological needs, which can impede calcium absorption and force the body to utilize medullary bone to meet its calcium requirements—potentially leading to conditions like osteoporosis (*Bains & Brake, 1995*). *Weiser et al. (1992)* demonstrated that vitamin C consumption significantly boosted the weight and strength of bones in broiler chickens.

Likewise, *Giang & Doan (1998)* found that vitamin C supplementation resulted in increased calcium and phosphorus levels in the tibia bones.

However, some studies present contrasting evidence. For instance, *Reyes et al. (2021)* reported that vitamin C supplementation up to 3,000 mg/kg did not influence egg quality or the mineral content in the tibiae of laying hens. In contrast, *Lohakare, Chae & Hahn (2004)* noted significant improvements in tibia strength linked to vitamin C supplementation in broilers compared to control groups.

In examining the interaction effects of vitamin $D_3$ and vitamin C, particularly at higher vitamin $D_3$ levels (5,500 IU) combined with additional vitamin C (150 mg), no significant enhancements in ash, calcium, or phosphorus levels in the carcasses of the offspring were found. However, an increase in dry matter content was observed, indicating a potential interaction that merits further exploration.

In conclusion, our results suggest that both vitamin $D_3$ and vitamin C play positive roles in the mineralization of broiler breeder offspring, primarily influencing calcium levels. While significant interactions between these vitamins in enhancing other mineral parameters were not evident, the increase in dry matter content with the combination of high levels of vitamin $D_3$ and vitamin C warrants further investigation.

### Effect of vitamin $D_3$ and vitamin C on blood biochemical parameters in broiler breeder hens

The impact of vitamin $D_3$ and vitamin C on the blood biochemical properties of broiler breeder hens is summarized in Table 4. Notably, higher levels of vitamin $D_3$ resulted in several significant effects, including a reduction in cholesterol, ALP, total antioxidant capacity, and PTH levels ($P < 0.05$). But triglyceride, glucose, AST, ALT, of broiler breeder hens blood serum were not affected by increased vitamin $D_3$. Although various mechanisms have been suggested to explain the effects of vitamin D on lipid profiles, its impact on blood lipid levels remains unclear. *Kim & Jeong (2019)* indicate that vitamin D may directly influence serum lipid profiles, such as triglycerides and cholesterol, by boosting bile salt production and lowering lecithin-cholesterol acyltransferase activity, as well as indirectly through enhanced calcium absorption, which leads to reduced fat absorption and increased hepatic bile acid synthesis from cholesterol. Similarly, *Dibaba (2019)* concluded that vitamin D supplementation appears to have a beneficial effect on reducing serum cholesterol and triglyceride levels. Concurrently, dos of vitamin $D_3$ increased calcium, progesterone, and $25(OH)D_3$ levels. The presence of $25(OH)D_3$ in serum serves as a critical marker for assessing vitamin $D_3$ status in the body. Our investigation confirmed that vitamin $D_3$ supplementation elevated $25(OH)D_3$ levels, aligning with findings from previous studies (*Wang et al., 2020*; *Zhang et al., 2020*; *Chen et al., 2020*; *Li et al., 2021*).

Vitamin $D_3$ is pivotal for enhancing dietary calcium absorption in the small intestine, playing a key role in calcium homeostasis. PTH is essential for regulating calcium ion concentrations in the bloodstream. Under conditions of low calcium, elevated PTH levels stimulate renal 1-alpha-hydroxylase, promoting the synthesis of $1,25(OH)_2D_3$. Conversely, increased extracellular calcium levels regulated by $1,25(OH)_2D_3$ inhibit PTH secretion (*Goltzman, 2018*). In our study, higher levels of vitamin $D_3$ (ranging from 3,500

**Table 4  The effect of vitamin $D_3$ and vitamin C on blood biochemical parameters in broiler breeder hens.**

| Treatments | | GLU[1] (mg/dL) | CHO[2] (mg/dL) | TRG[3] (mg/dL) | AST[4] (u/l) | ALT[5] (u/l) | ALP[6] (u/l) | CA[7] (%) | PHO[8] (%) | TAC[9] (mmol/dL) | UR[10] (mg/dL) | MDA[11] (μg/mL) | PRO[12] (pg/mL) | EST[13] (pg/mL) | 25OHD3 (ng/mL) | 1,25 OHD3 (pg/mL) | PTH[14] (pg/mL) | Calcitonin (pg/mL) |
|---|---|---|---|---|---|---|---|---|---|---|---|---|---|---|---|---|---|---|
| $D_3$ | 3,500 IU | 235.67 | 245.92a | 328.58 | 108.16 | 7.75 | 802.92a | 7.44b | 2.43 | 0.630a | 2.28 | 2.85 | 786.7b | 446.55 | 27.31b | 141.18 | 77.52a | 208.02 |
| | 5,500 IU | 231.33 | 185.50b | 328.25 | 114.58 | 9.66 | 704.17b | 9.12a | 2.36 | 0.558b | 1.76 | 2.94 | 1209.3a | 310.02 | 42.66a | 146.00 | 70.56b | 234.54 |
| *P-Value* | | 0.075 | 0.012 | 0.983 | 0.396 | 0.139 | 0.045 | 0.0001 | 0.701 | 0.039 | 0.123 | 0.845 | 0.022 | 0.087 | 0.0001 | 0.218 | 0.049 | 0.071 |
| SEM | | 9.78 | 15.49 | 11.03 | 5.23 | 0.880 | 32.68 | 0.252 | 0.121 | 0.022 | 0.229 | 0.328 | 121.23 | 53.66 | 2.10 | 3.11 | 2.35 | 9.85 |
| Vitamin C 0 mg | | 232.67 | 247.17a | 310.25b | 110.83 | 7.58 | 884.17a | 7.79b | 2.40 | 0.535b | 1.78 | 3.00 | 1012.09 | 301.98 | 33.45 | 129.45b | 77.77a | 206.47b |
| | 150 mg | 234.33 | 184.25b | 346.58a | 111.91 | 9.83 | 662.92b | 8.77a | 2.41 | 0.653a | 2.25 | 2.78 | 1178.50 | 454.58 | 36.53 | 157.73a | 70.32b | 236.08a |
| *P-Value* | | 0.905 | 0.009 | 0.030 | 0.885 | 0.085 | 0.0008 | 0.012 | 0.998 | 0.001 | 0.167 | 0.633 | 0.863 | 0.058 | 0.313 | 0.0001 | 0.036 | 0.046 |
| SEM | | 9.78 | 15.49 | 11.03 | 5.23 | 0.880 | 32.68 | 0.252 | 0.121 | 0.022 | 0.229 | 0.328 | 121.23 | 53.66 | 2.10 | 3.11 | 2.35 | 9.85 |
| Interaction | | | | | | | | | | | | | | | | | | |
| 3,500 $D_3$ × 0 C | | 249.33 | 298.83 | 316.50 | 105.83 | 6.50 | 914.16 | 6.96 | 2.36 | 0.536b | 2.08 | 3.25 | 802.10 | 349.26 | 26.16 | 126.80 | 82.33 | 196.42 |
| 3,500 $D_3$ × 150 C | | 222.00 | 193.00 | 340.66 | 110.50 | 9.00 | 691.66 | 7.91 | 2.50 | 0.723a | 2.47 | 2.45 | 771.25 | 543.83 | 28.46 | 155.56 | 72.71 | 219.61 |
| 5,500 $D_3$ × 0 C | | 216.00 | 195.50 | 304.00 | 115.83 | 8.66 | 774.16 | 8.61 | 2.43 | 0.533b | 1.49 | 2.76 | 1164.07 | 254.70 | 40.73 | 132.10 | 73.20 | 216.52 |
| 5,500 $D_3$ × 150 C | | 246.66 | 175.50 | 352.50 | 113.33 | 10.66 | 634.16 | 9.63 | 2.30 | 0.583b | 2.03 | 3.11 | 1254.56 | 365.33 | 44.60 | 159.90 | 67.92 | 252.55 |
| *P-Value* | | 0.051 | 0.064 | 0.444 | 0.633 | 0.842 | 0.382 | 0.926 | 0.445 | 0.048 | 0.819 | 0.230 | 0.727 | 0.586 | 0.795 | 0.913 | 0.521 | 0.649 |
| SEM | | 13.83 | 21.90 | 15.60 | 7.40 | 1.24 | 46.21 | 0.356 | 0.171 | 0.032 | 0.324 | 0.464 | 171.45 | 75.89 | 2.98 | 4.39 | 3.32 | 13.93 |

**Notes.**

[a-b] Means with different superscripts within a column differ significantly ($P < 0.05$).

SEM, Standard Error of the Mean; DM, Dry Matter.

[1] Glucose
[2] Cholesterol
[3] Triglyceride
[4] Aspartate amino transferase
[5] Alanine aminotransferase
[6] Akaline Phosphatase
[7] Calcium
[8] Phosphorus
[9] Total Antioxidant Capacity
[10] Uric Acid
[11] Malon Dialdehyde
[12] Progesterone
[13] Estrogen
[14] Parathyroid

to 5,500 IU) significantly decreased PTH concentrations, suggesting an increase in blood calcium levels. This finding contrasts with those reported by *Li et al. (2021)*.

The observed elevation in $1,25(OH)_2D_3$ levels aligns with findings from *Mazahery & von Hurst (2015)*, indicating that vitamin D supplementation is crucial for addressing deficiencies and maintaining adequate $25(OH)D_3$ levels. It is important to recognize that individual responses to vitamin $D_3$ supplementation may vary due to several factors, including baseline $25(OH)D_3$ levels, age, body composition, calcium intake, genetic background, and overall health status (*Mazahery & von Hurst, 2015*).

Our findings for vitamin C revealed a significant reduction in cholesterol, ALP and PTH levels, alongside increases in triglycerides, plasma calcium, TAC, and the active form of vitamin $D_3$ $(1,25(OH)_2D_3)$ ($P < 0.05$). However, vitamin C did not significantly affect other blood parameters. Supporting these results, *Aghajanian et al. (2015)* reported that vitamin C supplementation significantly decreased serum triglycerides and total cholesterol levels in aged mice compared to untreated controls. Vitamin C promotes the synthesis of $1,25(OH)_2D_3$, indirectly enhancing calcium mobilization from bones, which also influences serum calcium concentrations (*Demir et al., 1995*). Furthermore, ascorbic acid may amplify the genomic response to $1,25(OH)_2D_3$ in a vitamin D receptor-dependent manner, aiding in terminal differentiation and calcium regulation (*Lohakare et al., 2005*).

The combined effects of varying concentrations of vitamin $D_3$ and vitamin C significantly influenced the overall antioxidant capacity of the hens, with statistical significance observed ($P < 0.05$). Notably, low levels of vitamin $D_3$ supplemented with vitamin C produced the highest TAC. Significant interactions included an increase in TAC levels when 150 mg of vitamin C was combined with 3,500 IU of vitamin $D_3$. However, no significant interactions were found on other biochemical parameters in broiler breeder hens.

### Effect of vitamin $D_3$ and vitamin C on blood biochemical parameters of broiler breeder offspring

The influence of vitamin $D_3$ and vitamin C on the blood biochemical properties of broiler breeder offspring is summarized in Table 5. The findings indicate a significant effect of vitamin $D_3$ on plasma calcium, $25(OH)D_3$, PTH, and blood phosphorus levels ($P < 0.05$). Specifically, higher levels of vitamin $D_3$ resulted in increases in plasma calcium (11.45% *vs.* 9.54%), $25(OH)D_3$ (15.38 ng/mL *vs.* 10.21 ng/mL), and blood phosphorus (5.60% *vs.* 5.00%), which align with the results of *Khan et al. (2010)*.

*Farquharson et al. (1998)* suggested that ascorbic acid enhances the synthesis of $1,25(OH)_2D_3$, which correlates with the upregulation of the vitamin D receptor. In our study, higher levels of vitamin $D_3$ were associated with decreased concentrations of PTH. Elevated PTH levels typically respond to low blood calcium, particularly during demanding stages such as eggshell calcification (*Yamada et al., 2021*). PTH stimulates the resorption of medullary bone, leading to increased blood calcium levels. In calcium-deficient diets, laying hens show increased serum PTH, poorly calcified medullary bone, and decreased serum calcium and estrogen levels (*Yamada et al., 2021*). *Martins et al. (2017)* corroborate that low levels of $25(OH)D_3$ lead to decreased intestinal calcium absorption efficiency, stimulating PTH secretion. This presents a significant concern for older layer hens, which

**Table 5  The effects of vitamin D₃ and vitamin C on blood biochemical parameters of broiler breeder offspring.**

| Treatments | | Calcium (%) | 25(OH)D₃ (ng/mL) | 1,25(OH)₂D₃ (pg/mL) | PTH (pg/mL) | ALP (u/l) | Phosphorus (%) |
|---|---|---|---|---|---|---|---|
| D₃ | 3,500 IU | 9.54[b] | 10.21[b] | 69.08 | 70.09[a] | 2,206.50 | 5.60[a] |
| | 5,500 IU | 11.45[a] | 15.38[a] | 77.58 | 58.20[b] | 1,976.00 | 5.00[b] |
| *P-Value* | | 0.0001 | 0.0001 | 0.211 | 0.046 | 0.108 | 0.034 |
| SEM | | 0.268 | 0.711 | 4.65 | 3.96 | 97.01 | 0.187 |
| Vitamin C | 0 mg | 10.10 | 12.93 | 68.08 | 74.89[a] | 2,120.83 | 5.45 |
| | 150 mg | 10.88 | 12.66 | 78.58 | 53.40[b] | 2,061.7 | 5.15 |
| *P-Value* | | *0.054* | *0.793* | *0.126* | *0.001* | *0.670* | *0.270* |
| SEM | | 0.268 | 0.711 | 4.65 | 3.96 | 97.01 | 0.187 |
| Interaction | | | | | | | |
| | 3,500 D₃×0 C | 9.08 | 10.03 | 61.16 | 86.05 | 2,252.66 | 5.60 |
| | 3,500 D₃×150 C | 10.00 | 10.40 | 77.00 | 54.13 | 2,160.33 | 5.60 |
| | 5,500 D₃×0 C | 11.13 | 15.83 | 75.00 | 63.73 | 1,989.00 | 5.30 |
| | 5,500 D₃×150 C | 11.76 | 14.93 | 80.16 | 52.66 | 1,963.00 | 4.70 |
| SEM | | 0.379 | 1.00 | 6.58 | 5.60 | 137.19 | 0.264 |
| *P-Value* | | 0.713 | 0.536 | 0.427 | 0.077 | 0.811 | 0.270 |

**Notes.**

[a-b]Means with different superscripts within a column differ significantly ($P < 0.05$).

SEM, Standard Error of the Mean; TH, Parathyroid Hormone; ALP, Alkaline Phosphatase.

are more prone to fractures and mineralization defects. While vitamin D₃ supplementation effectively enhances serum 25(OH)D₃ levels and lowers PTH (*Chen et al., 2022*), our results indicated that the effects of vitamin D₃ on 1,25(OH)₂D₃ and ALP levels were not statistically significant. In contrast, vitamin C demonstrated a significant impact on PTH levels ($P < 0.05$). Specifically, the addition of 150 mg of vitamin C to the diet of broiler chickens resulted in a reduction of PTH levels in their offspring; however, no significant effects were observed on calcium, 25-hydroxycholecalciferol, ALP, or phosphorus levels.

The observed reduction in ALP activity in birds receiving vitamin C may be linked to decreased corticosteroid levels, which are known regulators of this enzyme (*Du et al., 2022*). Additionally, *Kucuk et al. (2003)* reported that vitamin C intake elevated blood calcium levels, which aligns with our findings. The combination of higher vitamin D₃ (5,500 IU) and vitamin C (150 mg) resulted in increased calcium levels, PTH, and 1,25(OH)₂D₃. The interaction effects were significant for PTH, as the combination of 5,500 IU D₃ and 150 mg vitamin C significantly lowered PTH compared to lower doses of D₃ and vitamin C. This indicates that optimal levels of both vitamin D₃ and C are essential for maintaining bone health and mineral balance in the offspring.

## CONCLUSION

This study demonstrates that dietary supplementation with vitamin D₃ and vitamin C positively influences bone mineralization, blood biochemical parameters, and overall health in broiler breeder hens and their offspring. The optimal dosages of 5,500 IU vitamin D₃ and 150 mg vitamin C significantly improved bone resistance, calcium content,

and antioxidant capacity in hens, while also promoting enhanced mineralization and bone health in their offspring. The interaction between these vitamins is crucial for boosting both bone strength and reproductive performance, providing valuable insights for optimizing dietary supplementation in poultry production. These findings contribute to the growing body of evidence supporting the importance of adequate vitamin $D_3$ and C intake for maintaining skeletal integrity and metabolic health in breeding hens and their progeny. However, the absence of a sham group in this study represents a limitation. This decision was made due to ethical considerations, as the study focused on minimizing unnecessary animal use and distress. Future studies could consider incorporating a sham group with careful ethical oversight to provide more robust comparative data. Future research should focus on identifying the optimal dosages and combinations of vitamin $D_3$ and vitamin C to maximize their benefits for bone health and overall poultry performance. Conducting long-term studies will be essential to assess the prolonged effects of vitamin supplementation on growth, health, and productivity in commercial poultry settings. Additionally, further investigation into how these vitamins influence calcium metabolism and bone health will provide deeper insights into their roles in poultry nutrition.

### Funding
The authors received no funding for this work.

### Competing Interests
The authors declare there are no competing interests.

### Author Contributions
- Ruhollah Kianfar conceived and designed the experiments, performed the experiments, prepared figures and/or tables, and approved the final draft.
- Reza Kanani conceived and designed the experiments, performed the experiments, prepared figures and/or tables, and approved the final draft.
- Hossein Janmohammadi conceived and designed the experiments, performed the experiments, prepared figures and/or tables, and approved the final draft.
- Majid Olyaee conceived and designed the experiments, performed the experiments, analyzed the data, prepared figures and/or tables, and approved the final draft.
- Maghsoud Besharati analyzed the data, authored or reviewed drafts of the article, and approved the final draft.
- Maximilian Lackner analyzed the data, authored or reviewed drafts of the article, and approved the final draft.

### Animal Ethics
The following information was supplied relating to ethical approvals (i.e., approving body and any reference numbers):

The Research Ethic Committees of University of Tabriz approved the study (IR.TABRIZU.REC.1403.008).

## Data Availability

The raw data are available in the Supplemental File.

## Supplemental Information

Supplemental information for this article can be found online at http://dx.doi.org/10.7717/peerj.18983#supplemental-information.

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
