# Peer review of "Implications of high-dose vitamin D₃ with and without vitamin C on bone mineralization and blood biochemical factors in broiler breeder hens and their offspring"

_PeerJ, doi:10.7717/peerj.18983_

## Round 0.1 · original submission · Major Revisions

Major revision is required by the authors. You can see that some very important concerns have been raised by one of the reviewers.

·

Basic reporting

No comment

Experimental design

No comment

Validity of the findings

No comment

Additional comments

Line 19-21 Abstract is supposed to be a summary, not to justify. I suggest it should be narrowed down to summary of results
Line 104 State the methods of Kanani et al 2023
Line 149-151 State the reference of the procedure used
Line 191 25-OHD3 has not been previously defined
Line 202 (P<0.05) define it
Line 297 Rephrase the sentence by constructing the grammar
Line 315 Conclusion should not include discussion. References should be excluded

·

Basic reporting

No comment

Experimental design

No comment

Validity of the findings

No comment

Additional comments

No comment

·

Basic reporting

A research problem statement should be included in the introduction.

The English language should be improved to ensure that an international audience can clearly understand your text.

The abstract is ambiguous and not clear. It should be rewritten to contain an brief into, aim, method, result and conclusion.

The introduction is ambiguous and not clear. The flow of the introduction can be improved for easy reading and information capturing. The context of the introduction can be improved tremendously in addition to referencing relevant paper not later tha the last 5 years. “Taking into account that calcium absorption decreases at the final stages of the egg production phase and vitamin D3 and C are effective factors on calcium absorption” is the justification for this study. Similarly, 2005 J. Appl. Poult. Res.14:670–678 had reported this effect on chick which will likely be the case for hens. What makes this work uniques?

The currency of the references need to be improved. Currently less than 18% (5 references) of the references in are from the last 5 years.

Experimental design

What informed the variation in Vitamin D administered concentration. A sham group without Vitamins D and C will be best to show if the changes might be as a result of Vitamins D and C supplementation.

The methods and the equipments need to be described in sufficient detail for replication.

Is there a reason why the biochemical parameters in the hens and chicks were different?

Statistical analysis should reflect only the statistical method(s) and the tool(s)used only.

No ethical statement in the manuscript.

Validity of the findings

Result reporting should be focused on the biomarkers assessed rather than speculating causes for the results eg Ln 191.

Results should be reported in an expanded form to show the effects of the vitamins C, D and their combination on each parameter with relevant disscussions where required.

Inclusion of measure of variability for all data and the n value is imperative.

Conclusions should be improved rather than have a repeat of the results. Future implications and research in addition to the limitations of this study should be included. Currently, the conclusions are not well stated, linked to the original research question and the supporting results are not in line with the result sections.
Result
Ln 168-170 and 187-190 sentence should be recast for clarity
Ln 169-170 should be expanded to show the effects of the vitamins C, D and their combination.
Ln 178-179 This statement negates the calcium result as the effects of the vitamins C, D and their combination had no effect on Ca.
Ln 188-190 should be expanded to show the effects of the vitamins C, D and their combination.
207-210 and 221-222 should be expanded to show the effects of the vitamins C, D and their combination.
Authors were silent about discussing Triglycerides increase by vitamin D
Ln 266-267 hematological parameters were not assessed. Kindly clarify

Conclusion
Ln 289 - 295 some of the results included in this section do not correspond with what was reported so crosscheck and recast.

Tables
Table header meaning should be described as footnotes for clarity.

Additional comments

This manuscript should be checked by a native English speaker.

---

## Round 0.2 · Minor Revisions

Dear Authors,
Please kindly attend to the concerns raised by the reviewer. You can see there is great progress, and these points raised are very important to improve the quality of your work.
Look forward to your revised manuscript.
Thank you very much

·

Basic reporting

The write up has been upgraded in addition to the currency of references.

Experimental design

The write up has been upgraded in addition to the currency of references. There are still some required adjustments.

Validity of the findings

The write up has been upgraded in addition to the currency of references.

Additional comments

Abstract
No mention of offspring in the methods please include.

Methods
Variation in Vitamin D administered should be backed by literature as pointed out by the authors in response to the first query.
The batch of the premix (vit. D and C), manufacturers and location whould be stated for reproducibility
3 in Vitamin D should be written as a subscript.
Include the manufacturer(s) of the ELISA kits and country of production.
The country where the AST and ALP kits were produced should be included.
ml should be mL.
There should be a space between numbers and SI units.
For biochemical analysis of the hens offsprings, the first sentence should be expanded to contain all procedure that was carried out for reproducibility. eg what was the animal state prior to blood collection, how old were the chicks, did they all have similar hatch time etc.

Result and Discussion
The numbers in the table signify which descriptive parameter? Likewise, there is an absence of variance in the data which should be included. This change should be reflected in text.
Authors are still silent about discussing Triglycerides increase by vitamin D
Ln. 303-307 has still not been recast in line with the results in the Table which should influence the discussion in 308-312. In addition, repitition of results should not be conflicted with discussion eg. Ln 310-312
In some section of results and discussion, some paragraphs are without references.

Conclusion
The absence of sham group should be included as a limitation and the reason should be stated as well.

Table 1: DCAB meaning should be included in the footnote

---

## Round 0.3 · accepted · Accept

I am delighted with the revised manuscript and agree with the reviewer it is acceptable for publication. Thank you authors for finding PeerJ as your journal of choice. I look forward to your future scholarly contributions. Congratulations :)

·

Basic reporting

No comment

Experimental design

No comment

Validity of the findings

No comment

Additional comments

No comment